# Adaptive IoU Thresholding for Improving Small Object Detection: A Proof-of-Concept Study of Hand Erosions Classification of Patients with Rheumatic Arthritis on X-ray Images

**DOI:** 10.3390/diagnostics13010104

**Published:** 2022-12-29

**Authors:** Karl Ludger Radke, Matthias Kors, Anja Müller-Lutz, Miriam Frenken, Lena Marie Wilms, Xenofon Baraliakos, Hans-Jörg Wittsack, Jörg H. W. Distler, Daniel B. Abrar, Gerald Antoch, Philipp Sewerin

**Affiliations:** 1Department of Diagnostic and Interventional Radiology, Medical Faculty, University Dusseldorf, D-40225 Dusseldorf, Germany; 2Rheumazentrum Ruhrgebiet, Ruhr-University Bochum, D-44649 Herne, Germany; 3Department of Rheumatology & Hiller Research Unit, University Hospital Dusseldorf, D-40225 Dusseldorf, Germany

**Keywords:** X-ray, imaging, deep learning, AI, RetinaNet, rheumatic arthritis, innovative techniques

## Abstract

In recent years, much research evaluating the radiographic destruction of finger joints in patients with rheumatoid arthritis (RA) using deep learning models was conducted. Unfortunately, most previous models were not clinically applicable due to the small object regions as well as the close spatial relationship. In recent years, a new network structure called RetinaNets, in combination with the focal loss function, proved reliable for detecting even small objects. Therefore, the study aimed to increase the recognition performance to a clinically valuable level by proposing an innovative approach with adaptive changes in intersection over union (*IoU*) values during training of Retina Networks using the focal loss error function. To this end, the erosion score was determined using the Sharp van der Heijde (*SvH*) metric on 300 conventional radiographs from 119 patients with RA. Subsequently, a standard RetinaNet with different *IoU* values as well as adaptively modified *IoU* values were trained and compared in terms of accuracy, mean average accuracy (*mAP*), and *IoU*. With the proposed approach of adaptive *IoU* values during training, erosion detection accuracy could be improved to 94% and an *mAP* of 0.81 ± 0.18. In contrast Retina networks with static *IoU* values achieved only an accuracy of 80% and an *mAP* of 0.43 ± 0.24. Thus, adaptive adjustment of *IoU* values during training is a simple and effective method to increase the recognition accuracy of small objects such as finger and wrist joints.

## 1. Introduction

Rheumatoid arthritis (*RA*) is a chronic inflammatory autoimmune disease that mainly affects the joints of the hands and feet and can lead to irreversible damage of the affected joints [1,2]. Therefore, early detection of erosions is essential, as drug treatment strategies can delay the potential joint destruction [3].

Due to its wide availability, relatively low cost, and high spatial resolution, conventional radiography, in particular, has emerged in recent decades as the essential tool for assessing RA stages and early diagnosis [2,4]. While imaging has become increasingly time-efficient due to technological advances, and biosensitive sequences such as Chemical Exchange Saturation Transfer (CEST) [5,6,7], delayed gadolinium-enhanced MRI of cartilage (dGEMRIC) [8], and T1rho have shown impressive performance in detecting bony lesions in current research [9], subsequent clinical assessment and documentation remain time-consuming and highly subjective.

Deep Learning (*DL*)-based algorithms have proven to be an excellent and timesaving alternative to human assessment [10,11]. Numerous studies have investigated and successfully validated the potential of DL for medical image classification and segmentation [10,11,12,13,14]. Some previous works have also explored these concepts for their suitability in assessing RA patients [13,15]. In numerous previous studies, detecting finger joints and erosions proved difficult and not yet clinically applicable [13,15,16]. Either they used two-stage approaches, such as cascade classifiers [16], in which joint localization and classification were developed separately, which may lead to errors, especially in destructed joints, or the models failed, especially in joint localization of the carpal bones, which could not be assessed separately because of their spatial distances [16]. Therefore, there is considerable scientific and clinical interest in new concepts for excellent classification of destructions and accurate and complete joint localization.

In the past two years, Retina networks (RetinaNet) [17], in particular, have emerged as a powerful tool for object detection in medical imaging and other fields [18,19,20]. Despite this, small object detection is still a challenging task [13,21], and many DL frameworks failed. Due to low resolution and background interference, negative areas (where no joint is shown) outweigh positive areas (where joint is shown) [22]. To alleviate this problem, Yan et al.’s study showed that the detection of small objects is especially poor when a high Intersection over Union (*IoU*) threshold is chosen. In contrast, a low *IoU* threshold leads to poor location accuracy [22]. However, in the field of RA research, there is considerable interest in small objects with close spatial relationships compared to other medical areas.

Therefore, the study aimed to investigate whether a novel approach based on adaptive *IoU* thresholds would provide better localization and detection accuracy. Our hypotheses were that (1) lower erosion detection thresholds in RA would result in higher detection performance, analogous to previous studies, and (2) that detection performance, as well as joint localization accuracy, could be substantially increased with a new adaptive IoU method.

## 2. Materials and Methods

### 2.1. Study Design

This is a retrospective image classification study conducted by local ethic regulations. The local ethics committee (Ethics Committee, Faculty of Medicine, Heinrich-Heine-University Düsseldorf, Germany, study number 2019-648) approved the study, and written consent was obtained for pseudonymized analysis in the context of clinic-related cases. To our knowledge, there is currently no open-source dataset of *RA* patients, including erosion classification, for standardized comparison with previous studies. For this reason, we crafted our own data set. Retrospectively, bilateral hand conventional radiographies of 119 patients (41 male, 78 female, age = 55.6 ± 12.2 years; age range = 23–87 years) with 300 radiographs acquired between 05/2005 and 04/2021 were analyzed for this purpose (Table 1). For all images, P.S., an experienced rheumatologist (15 years of experience in Rheumatology, including clinical trials and scoring Sharp van der Heijde (*SvH*) method), determined the erosion score according to van der Heijde et al. [23,24] assigning a value between 0 and 5 to each joint depending on the extent of erosion (Figure 1). In addition, the mean position of each joint was recorded. Furthermore, the size of each joint was measured using 20 patients, and based on this, bounding boxes were determined as ground truth (*GT*). Finally, the data set was strictly randomly divided into training (70%, number of patients = 83, number of images = 231), validation (10%; n = 12, number of images = 45), and test (20%, n = 24, number of images = 24) data. Initially, 24 of 37 patients with only one radiograph were assigned to the test group to test the models independently of repeated measurements. Subsequently, the remaining patients were randomly divided between training and test data, and one patient with all radiographs was assigned to the same group. To investigate substantial differences within a rater, P.S. rated the 24 images of the test data set again after six months.

### 2.2. Image Acquisition

All conventional radiographies (CR) were collected at the University Hospital Düsseldorf, Germany, with clinical x-ray systems. In all cases, the systems were from Siemens Healthineers (Siemens Healthineers, Erlangen, Germany), and only images showing both hands were used. In addition, all patients’ RA was diagnosed by an experienced rheumatologist. The images were downloaded from the in-house Picture Archiving and Communication System (PACS) and were available as Digital Imaging and Communications in Medicine (DICOM) 12-bit grayscale images. The image resolution ranged from 1572 × 1572 to 2738 × 2738 pixels.

### 2.3. Data Normalization

Based on the DICOM information window center and window width, the X-ray images were windowed and scaled in a range between 0 and 4096. Subsequently, the training data were modified using the subsequent data augmentation (Section 2.4). Finally, before the data were submitted to RetinaNet, all datasets were normalized based on Z-score normalization.

### 2.4. Data Augmentation

The performance of deep neural networks is directly related to the amount of data available for training [10,25]. Therefore, methods for systematic data augmentation and variation through transformations of the available images are essential. This study’s training data were artificially augmented using the open-source repository “imgaug” (https://github.com/aleju/imgaug, accessed on 1 February 2022). In medical research studies, as well as in our study, only a small number of patients are available. In order to generate a large amount of artificial data during training, extensive data augmentation was used. To this end, image contrast was varied during training using gamma adjustment (gamma = 0.5–2); images were shifted along the x and y axes by up to 5% of the image size, cropped from the border up to 5%, sometimes (*p* = 10%) rotate with 90 degrees, scaled with a factor between 0.5 and 2. In the end, images were rotated between −15 to 15 degrees (Figure 2).

### 2.5. Network Architecture

For simultaneous localization (i.e., joint classification) and erosion classification, a RetinaNet was used with a residual neural network (ResNet) as a bottom-up pathway (Figure 3a) to compress the global image information and a feature pyramid network (FPN) as a top-down pathway with skip connections to merge the top-down layers with the bottom-up layers (Figure 3b). Furthermore, one regressor network (Figure 3c) and two classification networks (Figure 3d,e), analogous to previous studies [18,20,26,27,28] were used. Here, the subnetworks were applied to the p2 and p3 outputs of the backbone (Figure 3).

All training runs were performed on a computer workstation with two Intel^®^Xeon^®^Gold 6242R Central Processing Units (Intel Corporation, Santa Clara, CA, USA), 376 GB of main memory, and four RTX 3090 Graphics Processing Units (NVIDIA, Santa Clara, CA, USA), with each model trained on only one GPU. The training and evaluation routines were implemented in Python (v3.8), and the model was implemented in PyTorch using PyTorch Lightning. ADAM optimizer to optimize the networks with an initial learning rate of 0.001 and a weight decay of 10^−6^ [29] was used. Additionally, a scheduler to adjust the learning rate during training (“ReduceLROnPlatau”, patience = 20, cooldown = 20 and factor = 0.1) was used. Networks were trained for 100 epochs based on the highest mean average accuracy (*mAP*) in the validation dataset; the optimal network was determined and applied to the test data. The focal loss function with alpha = 0.25 and gamma = 2 was used as the error function, analogous to previous publications [30,31]. In addition, the smooth l1 loss was used for the bounding box regression error calculation. Additionally, all images were resized to 400 × 400 pixels, and a batch size of 12 was used. Due to the small number of images and the wide-ranging data expansion, each image in the training dataset was predicted four times per epoch. Furthermore, in each block Batch normalization and as activation function Rectified Linear Unit (*ReLU*) were applied.

### 2.6. Adaptive IoU Threshold Fitting and Experimental Parameters

The thresholds for positive and negative regions are generally the same throughout the training. Especially for small objects, this has the consequence that the negative regions clearly predominate. Thus, the error function can be quickly and easily minimized by considering all regions as negative. This complicates further optimization since adjusting the weights will most likely result in false positive detection. Especially for the detection of small objects, it is, therefore, useful to change the *IoU* threshold during training (Equations (1) and (2)). At the beginning of the movement, it is more likely to achieve true positive detection by adjusting the weights and minimizing the error function. Subsequent gradual increases in the thresholds will achieve progressively more accurate localization.
(1)$$\mathrm{Intersection}\;\mathrm{over}\;\mathrm{Union}\;(\mathrm{IoU})=\frac{\mathrm{Area}\;\mathrm{of}\;\mathrm{Overlap}}{\mathrm{Area}\;\mathrm{of}\;\mathrm{Union}}$$
(2)$$current\;IoU=\frac{maximum\;IoU}{\Delta adaptiv\;epochs}current\;epoch$$

To validate the effectiveness of adaptive IoU thresholds for erosion classification of RA patients, the network architecture was trained with both fixed IoU thresholds and adaptive IoU thresholds. We varied both IoU-positive (0.5–0.3) and IoU-negative (0.4–0.2) entries and the delta (0.1–0.3) between these two points (Table 2). Adaptive IoU adjustments were applied over 50 and 100 epochs. This yielded 18 different network settings that were trained and validated separately.

### 2.7. Evaluation Metrics

To evaluate the different networks, *mAP* and accuracy were used, which are well-known and widely used standard measures for comparing models. Among these, *mAP* is the average value of AP across all object categories (Equation (3)). The higher the *mAP* value, the better the recognition performance, and vice versa. A joint was classified as correctly detected if, first, it was seen at the correct location (*IoU* ≥ 0.3 to the GT box) and, second, the correct erosion value was detected. In all other cases, the detection was classified as false. In addition, the mean *IoU* value was determined since a high match is associated with high positional accuracy.
(3)$$mean\;average\;accuracy\;\left(mAP\right)=\frac{1}{\left|classes\right|}\sum_{c\;\in \;classes}\frac{\left|TP_{c}\right|}{\left|FP_{c}\right|+\left|TP_{c}\right|\;}$$

### 2.8. Statistical Analyses

Statistical analyses were performed by K.L.R. in Python by using the “SciPy” (version 1.9.1) Open-Source environment. The Kendall-Tau rank correlation coefficient (τ) was determined to investigate possible statistical relationships between erosion scores and sum erosion scores assigned by the model, and those by the rheumatologist were assessed using Kendall-Tau (τ) rank correlation coefficients. The tau effect size was classified as low (0.1–0.3), medium (0.3–0.5), and strong (>0.5), according to Cohen et al. [32].

Due to the experimental design of this study, the significance level was set from *p* ≤ 0.05 to an adjusted *p* ≤ 0.0083 according to the conservative alpha adjustment method Bonferroni [33]. This “low” significance level prevented alpha error inflation while maintaining statistical power.

## 3. Results

### 3.1. Patient Characteristics

After the random assignment of patients to the three data sets (training, validation, and test), the demographic and medical data of the groups were similar (Table 2). The test patients had a mean age of 57 ± 12 years (27–87) and a mean sum erosion score of 55 ± 30 (32–179). The different disease stages of the SvH score are described in Table 3 and listed for the test patient group, including the relative frequency. All assessments were performed by the same rheumatologist who was specifically trained for this task.

### 3.2. Detection Accuracy and Localization Accuracy

The agreement of the automatically and manually detected joint position and erosion score was assessed in the test group patients using the *mAP* and accuracy, and the mean *IoU* was determined. Similar to previous studies, we found that decreasing *IoU* thresholds increased accuracy and *mAP* for the static approach (Table 4). In comparison, the adaptive approach allowed higher *IoU* thresholds, which favored higher site accuracy (Table 5). In addition, accuracy and *mAP* were significantly higher than without adaptive adjustment, at 0.95 ± 0.04 and 0.81 ± 0.15, respectively. This is also consistent with the graphically observed results and the perception of the rheumatologist who visually reviewed the predictions (Figure 4). Localization accuracy scaled and depended on the number of adaptive epochs, primarily with the *IoU* pos threshold. Analogous to the detection accuracy, we observed higher joint localization accuracy using the adaptive approach (Table 5).

### 3.3. Agreement and Differences between Automatic and Manual Evaluation

For the best-trained RetinaNet, a high degree of agreement was found between the automatic erosion classification and the manual classification, with an accuracy of 94% and an mAP of 0.81. Repeatability within the reader was also found, with an accuracy of 88.5% and an mAP of 0.79. However, differences were observed in the distribution of deviations between the automatic and both manual classifications. For the distribution of prediction (DL vs. manual time point 1, Figure 5a) or reproducibility (manual time point 1 vs. time point 2 after 6 months, Figure 5b), we observed differences as a function of classes. The diagonal elements of the confusion matrix represent the number of points where the predicted or reproduced prediction matched the true label (manual time 1, GT), while the off-diagonal elements are those that were mislabeled by the classifier. The higher the diagonal values of the confusion matrix, the better, as this indicates many correct predictions. Due to the class imbalance, all data were normalized (to the number of elements in each class). While the rheumatologic assessment differed only by an erosion score, the DL approach classified joints with an erosion score of 0 more frequently (Figure 5). In addition, the lowest class accuracy of the DL framework, 0.48, was significantly lower than the human assessment by an experienced rheumatologist, 0.64.

Moreover, for the DL-based SvH assessment, we found a strong and significant correlation for both the individual joints (τ = 0.92, *p* < 0.001) and the sum score (τ = 0.88, *p* < 0.001). Similarly, we observed a strong and significant correlation between the two human assessment time points for the individual joints (τ = 0.90, *p* < 0.001) and the sum score (τ = 0.92, *p* < 0.001). This is also illustrated by the covariance matrix shown in Figure 4.

### 3.4. Time Required

On average, DL-based analysis of an X-ray image took less than 1 s on the specialized workstation and 5 s on a standard clinical workstation. Both are substantial time savings compared to evaluation by an experienced rheumatologist, who is estimated to take 9 ± 13 min (measured duration of P.S. for 20 randomly selected patients) to evaluate and document each finger and wrist joint.

## 4. Discussion

In this study, we successfully presented a new method to substantially improve the recognition accuracy of small objects by adaptively adjusting the *IoU* thresholds during the training process.

In recent years, the research field of artificial intelligence has expanded in all areas of medicine, including radiology and rheumatology [11]. Especially in image processing, innovative techniques have shown great promise in improving and speeding up clinical workflows and reducing the workload of medical staff. Given the clinical experience, requirements, and human need to monitor DL pipeline decisions, we decided to implement joint erosion classification using a RetinaNet. Compared to feedforward neural networks (FNN) that only have a vector of classifications at the end of the convolutions, for example, to decide if the patient has corona on the CT image or not [14], the RetinaNet allows a visual representation of which region the respective decision is based on. The radiologist or rheumatologist is thus able to verify and confirm the results.

Although joint damage is increasingly assessed with echography and MRI examinations, radiographs still provide a comprehensive or panoramic view of joints and are the clinical standard for classifying RA stages [16]. Deep-learning algorithms could greatly improve the clinical assessment of radiographs in many cases, e.g., Pneumonia detection in chest X-ray images or bone lesion detection in musculoskeletal radiographs [34,35]. In addition, new radiographic findings of joint destruction could be discovered. Recently, numerous studies have reported that Deep Learning or CNN was used to assess joints or bones. In this regard, different types of osteoarthritis have been investigated, including osteoarthritis of the hip [36], and osteoporosis of the knees [37], but also the assessment of bone age [38] has been the focus of the studies. However, these previous studies considering large joints have limited applicability to RA as a polyarthritis with central involvement of small joints. Our study overcomes the difficult task of identifying small joints, thus closing the gap in RA joint classification.

In our study, we observed a dependence between detection accuracy and the *IoU* thresholds used, analogous to Yan et al. [22]. Furthermore, no trained RetinaNet with fixed *IoU* values over the training epochs achieved sufficient accuracy for clinical applicability.

In particular, many joints of different sizes are located close to each other in the carpal region. Analogous to the study of Hirano et al., who achieved a localization accuracy of 95.3% for the finger joints using a two-stage approach [16], which is comparable to our study, the intercarpal joints tended to be neglected in their study because these areas are complex and have a closer spatial relationship. We observed similar results with single-stage RetinaNet without adaptive adjustment of the *IoU* threshold. However, the final model with adaptive *IoU* adjustment was able to capture complex regions by adaptive adjustment during training. Therefore, it can be assumed that adaptive adaptation is not only suitable for small objects but also of interest for complex structures with close spatial relationships.

Due to the lack of localization and low accuracy in erosion detection, none of the models we tested without adaptive *IoU* values achieved sufficient accuracy for routine clinical use. In comparison, with the proposed adaptive approach and end-Pos-IoU\end-Neg-IoU values of 0.4\0.3 and 50 adaptive epochs or end-Pos-IoU\end-Neg-IoU values of 0.5\0.3 for 100 adaptive epochs, an accuracy of more than 94% was achieved with an mAP of 0.81 ± 0.18 (50 adaptive epochs) and an *mAP* of 0.79 ± 0.22 (100 adaptive epochs). These results are comparable to the repeatability of an experienced rheumatologist (*mAP* = 0.79 and accuracy 88.5%) within one evaluator.

Similar results were observed by Wang et al. in their study on JSN classification in patients with RA [22]. They achieved only a maximum mAP of 0.71 using the classical You only look once (YOLO) version 4 approach. Their proposed adjustment of error functions based on the distance to the GT box, loss generalization, consideration of aspect ratios, and separation of hand and finger joints increased the performance to mAP = 0.87 for two-hand radiographs. However, the performance was determined using validation data rather than test data, and that hands in advanced stages of degeneration were excluded, making a comparison difficult. Nonetheless, our proposed fit is straightforward, requires no prior assumptions, and is extensible to any data set. In addition, we could classify both wrists and finger joints simultaneously in models without requiring additional computational steps. This significantly reduces the computational power and, thus, the availability of usable hardware in the clinical setting. On a standard workstation without GPU used in clinic, a complete evaluation requires only 5 s, which allows a significant acceleration of the clinical routine.

Our study, as well as the study by Wang et al., impressively show that the recognition accuracy can be significantly improved by adjusting the loss as a function of spatial relationship compared to previous RA studies, in which only recognition accuracies between 70.6 to 77.5 could be achieved [16,39], we targeted mAP values of 0.81 and 0.87, respectively.

In addition, it was notable that the best model we trained had a higher agreement with the rater than the rater had with himself at a delay of six months. This could be because the model generalized the subjective decision-making of the rater for the first time. Nevertheless, the RetinaNet sometimes differed from the rheumatologist by more than one SvH score. In contrast, at six months, the rheumatologist differed from his previous evaluation by no more than one score. Here, the model tended to classify joints with a score of 1–4 as score 0, which could be due to the class imbalance of the individual scores. While score 0 was present in 75.26% of the joints, scores 1–4 were present in only 3.13–6.9% of the joints.

Furthermore, our study shows that the medical care of RA patients can be optimized in terms of time by using deep learning frameworks. Experienced rheumatologists need 9 ± 13 min for a complete erosion history and documentation. In contrast, the RetinaNet we used took about 5 s for equivalent documentation. This time savings could allow physicians to spend less time on documentation alone in the years to come. In this way, rheumatologists can spend more time with their patients and perform tasks, such as face-to-face discussions with patients about clinical problems and limitations, that cannot be performed equivalently by DL frameworks. Nevertheless, some limitations have to be mentioned. First, the number of patients was limited, mainly due to the fact that there was no freely available data set. Consequently, we need to prepare our own dataset, which is time-consuming work. Second, we only examined CR from Siemens Healthineers. However, compared to MRI measurements in which numerous different scanner coil configurations are available, X-ray images, on the other hand, can be considered comparatively uniform. However, this study did not consider the effects of variability between different providers, platforms, and institutes. In addition, the effects of rings or other interfering objects on the accuracy of the assessment were not investigated. Therefore, further studies are needed to validate its applicability across multiple institutions and other x-ray manufacturers. Third, our proposed approach was only studied for one retinal network. Although retinal networks have been shown in numerous studies to have higher accuracy compared to other network configurations such as YOLO, single-shot multi-box detectors (SSD), etc. [18,40], further studies are needed to investigate the benefits of adaptive adjustment of *IoU* thresholds as well as to evaluate the different model types in assessing erosion values. Fourth, the paragraph we gave was applied exclusively to images in which all objects were small but of comparable size. Its usefulness for classification tasks in which objects of different sizes are to be detected must therefore be investigated in subsequent studies.

## 5. Conclusions

In our study on small object detection using erosion classification of RA patients as an example, we successfully validated a new approach to adaptive *IoU* values. Our proposed method makes it possible to determine joint erosions on hand radiographs with high accuracy, which potentially provides a reliable and fast methodology to improve clinical workflow.

## Figures and Tables

**Figure 1 diagnostics-13-00104-f001:**
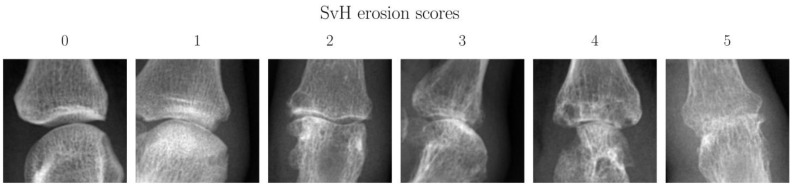
Representative conventional radiographs (CR) of finger joints as a function of damage, classified according to the Sharp van der Heijde (SvH) method by an experienced rheumatologist. From left to right, finger joints are sorted by SvH score from normal (SvH score = 0) to severely damaged (SvH score = 5) joint. Above each image, the respective score is indicated.

**Figure 2 diagnostics-13-00104-f002:**
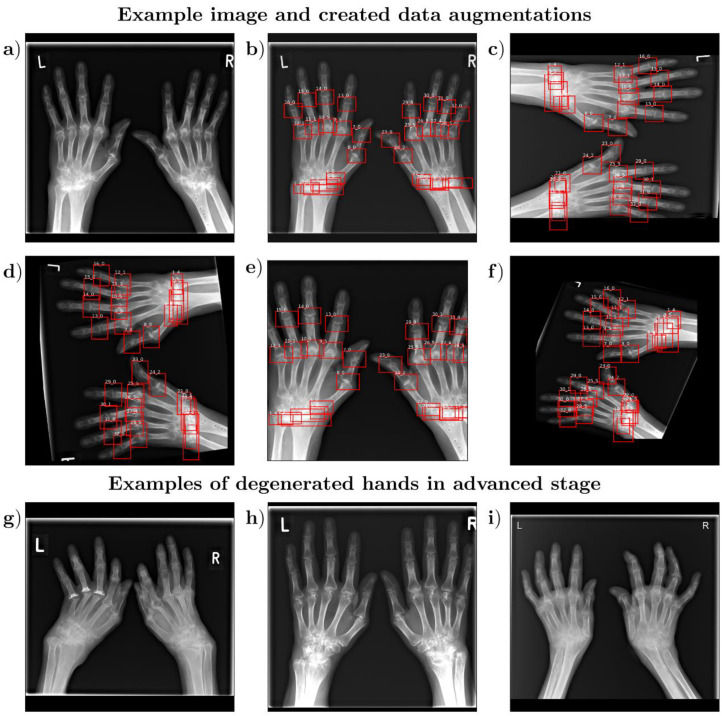
Representative visualization of a hand and its data expansion, as well as example images for hands in the degenerated stage. Shown are the original radiograph without the bounding boxes (**a**) as well as with the bounding boxes (**b**) and four exemplary baseline examples (**c**–**f**) after data expansion, including the marked bounding boxes of P.S. (rheumatologist with 15 years of experience). Two numbers are indicated above each bounding box, separated by an underscore. Here, the first number represents the number of the joint (1–32), and the second represents the specific erosion value. Furthermore, three hands in the advanced stage of degeneration (**g**–**i**) are shown, which were in the test data set to allow a complete evaluation of the accuracy of the model.

**Figure 3 diagnostics-13-00104-f003:**
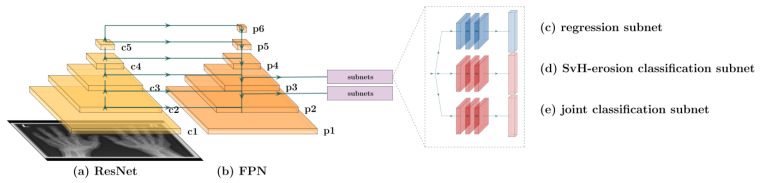
Schematic representation of the RetinaNet, which was used to detect and classify X-ray hand images automatically. The residual neural network (ResNet) (**a**) compressed the global image information. Then, the compressed data are merged by local skip connections (blue arrows) and processed in the feature pyramid network (FPN) (**b**). The regression subnetwork (**c**) and the two classification subnetworks (**d**,**e**) were then applied to outputs p2 and p3. Furthermore, batch normalization and Rectified Linear Unit (ReLU) were used as the activation function.

**Figure 4 diagnostics-13-00104-f004:**
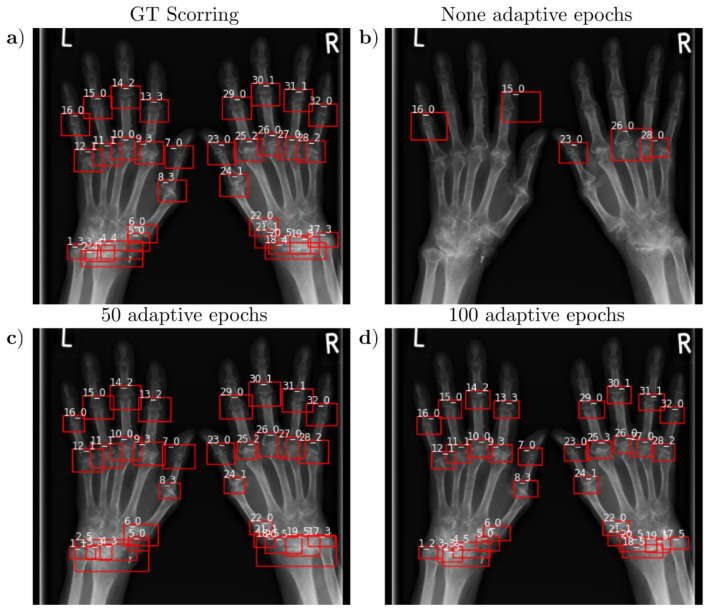
Visualization of a representative hand of a 68-year-old female patient and SvH erosion classifications for the best-trained models according to the mAP metric of the validation patients as a function of adaptive epochs and the corresponding manual reference classifications from an experienced rheumatologist. (**a**) Reference ground truth (GT) image from an experienced rheumatologist. (**b**) Without adaptive adjustment of IoU values during training, not all joints were localized, and the network mainly classified joints with an erosion value of 0, regardless of joint degeneration. Retinal networks trained with adaptive adaptation over 50 epochs and IoU pos\neg 0.4\0.3 values (**c**) or 100 epochs and IoU pos\neg 0.5\0.3 values (**d**) recognized all joints. Strikingly, adaptation over 100 epochs provided slightly higher localization accuracy and outperformed GT localization accuracy. In contrast, in the model fitted over 50 epochs, all erosion classifications matched the reference image, whereas the model held over 100 adaptive epochs misclassified two joints.

**Figure 5 diagnostics-13-00104-f005:**
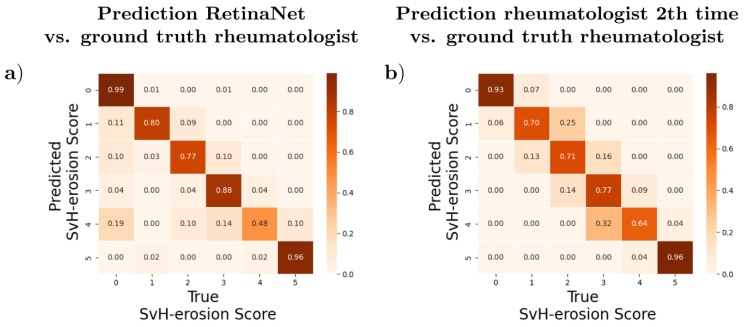
Comparison of confusion matrices: the X axis shows the assumed ground truth, i.e., the SvH erosion score of the experienced rheumatologist at time point 1, and the Y axis represents the erosion scores predicted by RetinaNet (**a**) and the erosion scores at the second time point determined by the same experienced rheumatologist, respectively (**b**). We found that both the AI and the rheumatologist confidently classified the extreme points SvH erosions with a score of 0 and 5, respectively, and showed no discrepancies. However, RetinaNet tended to classify a joint as 0 more often than average, whereas the rheumatologist deviated exclusively by one from the previous score.

**Table 1 diagnostics-13-00104-t001:** Overview of the demographic and medical characteristics of the patients as function of group. All values are shown as mean ± standard deviation; in brackets, the range is additionally given if useful. If more than one X-ray image were available for a patient, the age and scoring at the time of the first X-ray were used.

Parameters	Overall	Training	Validation	Test
Age [a]	55.6 ± 12.2 (23–87)	55 ± 13 (23–87)	56 ± 7 (29–82)	57 ± 12 (27–87)
Number patients	119	83	12	24
Number images	300	231	45	24
male/female	41/78	29/54	4/8	6/18
Sum of erosion score	52 ± 22	53 ± 25 (32–179)	46 ± 8 (32–101)	55 ± 30 (32–179)
Mean erosion score	1.52 ± 0.70 (1.0–5.59)	1.65 ± 0.77 (1.0–5.59)	1.43 ± 0.25 (1.0–4.3)	1.72 ± 0.95 (1–5.59)
Min erosion score	0.0 ± 0.0 (0–0)	0.0 ± 0.0 (0–0)	0.0 ± 0.0 (0–0)	0.0 ± 0.0 (0–0)
Max erosion score	4.1 ± 1.92 (1–6)	4.2 ± 1.87 (1–6)	3.2 ± 1.7 (1–6)	4.6 ± 1.6 (1–6)

**Table 2 diagnostics-13-00104-t002:** Overview of the 18 network configurations investigated. Each *IoU* positive\negative threshold pair was trained for the static case (None) or with an adaptive adjustment of 50 or 100 epochs.

RetinaNet Number	*IoU* Positive Threshold	*IoU* Negative Threshold	Adaptive Epochs
1–3	0.5	0.4	None, 50, 100
3–6	0.4	0.3	None, 50, 100
7–9	0.3	0.2	None, 50, 100
10–12	0.5	0.3	None, 50, 100
13–15	0.4	0.2	None, 50, 100
16–18	0.5	0.2	None, 50, 100

Abbreviation: IoU—Intersection over Union, None—No adaptive adjustment.

**Table 3 diagnostics-13-00104-t003:** Overview of the distribution of the disease stages according to the SvH erosion classification based on the 24 patients and 768 joints in the test data set [24].

SvH Erosion Score	Description	Number of Joints	Proportion
0	Normal joint	578	75.26%
1	discrete erosion	53	6.90%
2	Large erosion not passing midline *	35	4.56%
3	Large erosion passing midline *	27	3.52%
4	Sum of combined scores equals four	24	3.13%
5	Sum of combined scores equal to or larger than five	51	6.64%

Abbreviation: *—Previous points can be combined into 2 or 3.

**Table 4 diagnostics-13-00104-t004:** Detection accuracy and map of the studied models as a function of the threshold and the number of adaptive epochs. Each value is expressed as mean and standard deviation.

Pos/Neg *IoU* Threshold	None		50 Adaptive Epochs		100 Adaptive Epochs	
	Accuracy	*mAP*	Accuracy	*mAP*	Accuracy	*mAP*
0.5/0.4	0.00 ± 0.00	0.00 ± 0.00	0.79 ± 0.11	0.45 ± 0.26	0.77 ± 0.15	0.40 ± 0.23
0.4/0.3	0.15 ± 0.07	0.25 ± 0.19	0.94 ± 0.05	0.81 ± 0.18	0.57 ± 0.18	0.27 ± 0.19
0.3/0.2	0.80 ± 0.14	0.43 ± 0.24	0.92 ± 0.06	0.67 ± 0.23	0.90 ± 0.08	0.65 ± 0.23
0.5/0.3	0.00 ± 0.00	0.00 ± 0.00	0.65 ± 0.20	0.28 ± 0.19	0.94 ± 0.06	0.79 ± 0.22
0.4/0.2	0.07 ± 0.45	0.21 ± 0.14	0.56 ± 0.17	0.27 ± 0.17	0.65 ± 0.17	0.26 ± 0.18
0.5/0.2	0.20 ± 0.08	0.27 ± 0.19	0.42 ± 0.13	0.27 ± 0.19	0.79 ± 0.15	0.34 ± 0.22

Abbreviation: pos—positive; neg—negative; IoU—Intersection over Union; mAP—mean average accuracy, None—No adaptive adjustment.

**Table 5 diagnostics-13-00104-t005:** Location accuracy of the studied RetinaNet models as a function of the threshold and the number of adaptive epochs. Each value is expressed as mean and standard deviation.

Pos/Neg *IoU* Threshold	None	Adaptive Epochs	
		50	100
0.5/0.4	0.00 ± 0.00	0.59 ± 0.19	0.6 ± 0.14
0.4/0.3	0.11 ± 0.24	0.72 ± 0.14	0.45 ± 0.28
0.3/0.2	0.63 ± 0.12	0.68 ± 0.12	0.68 ± 0.16
0.5/0.3	0.00 ± 0.00	0.48 ± 0.25	0.65 ± 0.07
0.4/0.2	0.05 ± 0.16	0.44 ± 0.27	0.47 ± 0.21
0.5/0.2	0.20 ± 0.09	0.33 ± 0.30	0.59 ± 0.15

Abbreviation: pos—positive; neg—negative; IoU—Intersection over Union; None—No adaptive adjustment.

## Data Availability

Data and evaluation scripts can be provided by the authors upon reasonable request.

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
