# Peer review of "Adaptive IoU Thresholding for Improving Small Object Detection: A Proof-of-Concept Study of Hand Erosions Classification of Patients with Rheumatic Arthritis on X-ray Images"

_diagnostics, 2022, doi:10.3390/diagnostics13010104_

Round 1
Reviewer 1 Report
1 you retrospectively analyzed 300 conventional radiographs from 119 RA patients and determined the erosion score. how many radiographs did each patients have ?also in table 2 ,in test group ,24 patients included 24 radiograph mean one radiograph for each patients ,is there any statistic bias for test group ?why did not you select radiograph randomly for each group?please explain it .
2 regarding RA disease ,there are many type ,have you consider different type of radiographs will infuence the result?
3 it is better to add a table to show RA patients‘ characteritics.
4 In discussion ,have you compare the accuracy with oher study ?or have you evaluate the accuracy by radiograph experts?
Author Response
Title: Adaptive IoU thresholding for improving small object detection: A proof of concept study of hand erosions classification of patients with rheumatic arthritis on X-ray images
Manuscript ID: diagnostics-1990352
Journal: Diagnostics
Responses to the Reviewers’ Comments
The authors thank the reviewers for their careful revision of the manuscript and valuable comments, which are addressed and considered in the revised version of the manuscript. Please note that all changes made to the manuscript have been highlighted using the “Track changes”-mode in Microsoft Word and are detailed in this document, too. Please note that line numbers refer to the main text body of the revised document.
Replay to Reviewer 1
R1 Comment#1: “you retrospectively analyzed 300 conventional radiographs from 119 RA patients and determined the erosion score. how many radiographs did each patients have? also in table 2, in test group ,24 patients included 24 radiographs mean one radiograph for each patient, is there any statistic bias for test group? why did not you select radiograph randomly for each group? please explain it.”
Authors’ Response: We agreed with the reviewer that the distribution of patients and the number of images were not adequately described in the manuscript. Due to the retrospective nature of the study, we had a different number of radiographs for each patient (between 1 and 5). In order to create an independent test group from the training and validation group, we included only patients with only one radiograph in the test group (there were 37 patients with just one radiograph). According to Table 1, there were no statistical differences between the groups in terms of the medical parameters (as indicated in Table 1).
Authors‘ Action: Based on the reviewer's excellent commentary, we explicitly describe and justify the distribution of patients among the three groups. The revised section now reads:
„Finally, the data set was strictly randomly divided into training (70%, number of patients = 83, number of images = 231), validation (10%; n = 12, number of images = 45), and test (20%, n = 24, number of images = 24) data. Initially, 24 of 37 patients with only one radiograph were assigned to the test group to test the models independently of repeated measurements. Subsequently, the remaining patients were randomly divided between training and test data, and one patient with all radiographs was assigned to the same group. To investigate substantial differences within a rater, P.S. rated the 24 images of the test data set again after six months. “
(lines 100ff)
R1 Comment#2: “regarding RA disease, there are many types, have you considered different type of radiographs will influence the result?”
Authors’ Response: We thank the reviewer for this further helpful comment. Rheumatoid arthritis has different disease courses, but SvH scoring is a universal measure in this regard, defined independently of RA type. Compared with other imaging modalities such as MRI, radiographs are clearly standardized. Thus, radiographs from different institutes and from different radiographic devices were used in our study, but it is undisputed that the quality of the image affects the accuracy of the scoring.
Authors‘ Action: Considering this important point of the reviewer and the fact that image quality always has an impact on neural network performance, we have elaborated on this point in the Limitations section of the Discussion. The corresponding section in the limitation reads:
“Second, we only examined CR from Siemens Healthineers. However, compared to MRI measurements in which numerous different scanner coil configurations are available, X-ray images, on the other hand, can be considered comparatively uniform. However, this study did not consider the effects of variability between different providers, platforms, and institutes. In addition, the effects of rings or other interfering objects on the accuracy of the assessment were not investigated. Therefore, further studies are needed to validate its applicability across multiple institutions and other x-ray manufacturers.” (line 419ff)
R1 Comment#3: “it is better to add a table to show RA patients‘ characteristics.”
Authors’ Response and Action: We agreed with the reviewer that a table helps to show the characteristics of the patients. In agreement with the reviewer, we have moved the previous Table 2 (Patient characteristics) from the Results section to the Material & Methods section and added a column summarizing Overall patient characteristics. The revised table reads:
Table 1. Overview of the demographic and medical characteristics of the patients as function of group. All values are shown as mean ± standard deviation; in brackets, the range is additionally given if useful. If more than one X-ray image were available for a patient, the age and scoring at the time of the first X-ray were used.
|
parameters |
Overall |
Training |
Validation |
Test |
|
Age [a] |
55.6 ± 12.2 (23 – 87) |
55 ± 13 (23 – 87) |
56 ± 7 (29 – 82) |
57 ± 12 (27 – 87) |
|
Number patients |
119 |
83 |
12 |
24 |
|
Number images |
300 |
231 |
45 |
24 |
|
male / female |
41 / 78 |
29 / 54 |
4 / 8 |
6 / 18 |
|
Sum of erosion score |
52 ± 22 |
53 ± 25 (32 – 179) |
46 ± 8 (32 – 101) |
55 ± 30 (32 – 179) |
|
Mean erosion score |
1.52 ± 0.70 (1.0 – 5.59) |
1.65 ± 0.77 (1.0 – 5.59) |
1.43 ± 0.25 (1.0 – 4.3) |
1.72 ± 0.95 (1 – 5.59) |
|
Min erosion score |
0.0 ± 0.0 (0 - 0) |
0.0 ± 0.0 (0 - 0) |
0.0 ± 0.0 (0 – 0) |
0.0 ± 0.0 (0-0) |
|
Max erosion score |
4.1 ± 1.92 (1 – 6) |
4.2 ± 1.87 (1 – 6) |
3.2 ± 1.7 (1 – 6) |
4.6 ± 1.6 (1 – 6) |
R1 Comment#4: “In discussion, have you compared the accuracy with other study? or have you evaluated the accuracy by radiograph experts?”
Authors’ Response: We thank the reviewer for this critical inquiry. In our study, we determined the accuracy of our model based on the scoring of the expert radiographer and compared it with other studies.
Authors‘ Action: In light of the questions, we have revised the wording in the Discussion section in which we compared our accuracy to accuracy in other studies for better readability. The revised section reads:
“Due to the lack of localization and low accuracy in erosion detection, none of the models we tested without adaptive IoU values achieved sufficient accuracy for routine clinical use. In comparison, with proposed adaptive approach and end-Pos-IoU \ end-Neg-IoU values of 0.4 \ 0.3 and 50 adaptive epochs or end-Pos-IoU \ end-Neg-IoU values of 0.5 \ 0.3 for 100 adaptive epochs, an accuracy of more than 94% was achieved with an mAP of 0.81 ± 0.18 (50 adaptive epochs) andan mAP of 0.79 ± 0.22 (100 adaptive epochs). These results are comparable to the repeatability of an experienced rheumatologist (mAP = 0.79 and accuracy 88.5%) within one evaluator. Similar results were observed by Wang et al. in their study on JSN classification in patients with RA [23]. They achieved only a maximum mAP of 0.71 using the classical You only look once (YOLO) version 4 approach. Their proposed adjustment of error functions based on the distance to the GT box, loss generalization, consideration of aspect ratios, and separation of hand and finger joints increased the performance to mAP = 0.87 for two-hand radiographs.” (line 373ff)
Reviewer 2 Report
In recent years, much research evaluating the radiographic destruction of finger joints in patients with rheumatoid arthritis (RA) using deep learning models was conducted. Unfortunately, most previous models were not clinically applicable due to the small object regions as well as the close spatial relationship. In recent years, a new network structure called RetinaNets, in combination with the Focalloss function, proved reliable for detecting even small objects.
Based on this, the authors aimed to increase the recognition performance to a clinically valuable level by proposing an innovative approach with adaptive changes in intersection over union (IoU) values during training of our Retina Networks using the focal loss error function.
To this end, the authors retrospectively analyzed 300 conventional radiographs from 119 RA patients and determined the erosion score according to the Sharp van der Heijde (SvH) metric.
Subsequently, they trained a standard RetinaNet with different IoU values as well as adaptively modified IoU values and compared them in terms of accuracy, mean average accuracy (mAP), and IoU.
The authors demonstrated that their approach, improved the erosion detection accuracy up to 94% and an mAP of 0.81 ± 0.18. In contrast with static IoU values, they only achieved an accuracy of 80% and an mAP of 0.43 ± 0.24.
They concluded that the adaptive adjustment of IoU values during training is a simple and effective method to increase the recognition accuracy of small objects such as finger and wrist joints.
Innovative, interesting and well written manuscript.
I have some minor suggestions with a pure academic spirit:
1. I suggest to be more explicit in the purpose “Therefore, our study aimed to investigate how the IoU threshold affects classification accuracy and whether a novel approach we proposed using adaptive IoU thresholds would provide higher detection and localization accuracy.”
2. Describe figure 1 in the body of the manuscript. Furthermore, the legend must explain the six parts of the figure.
3. Check the resolution of the figure.
4. What about inserting a table/list with the acronyms?
5. Avoid the use of we/our
6. “29 \ 54” Usually the slash is “/”
7. Check if equations (1-3) follow the MDPI standards
Author Response
Title: Adaptive IoU thresholding for improving small object detection: A proof of concept study of hand erosions classification of patients with rheumatic arthritis on X-ray images
Manuscript ID: diagnostics-1990352
Journal: Diagnostics
Responses to the Reviewers’ Comments
The authors thank the reviewers for their careful revision of the manuscript and valuable comments, which are addressed and considered in the revised version of the manuscript. Please note that all changes made to the manuscript have been highlighted using the “Track changes”-mode in Microsoft Word and are detailed in this document, too. Please note that line numbers refer to the main text body of the revised document.
Replay to Reviewer 2
R2 General Comment: “In recent years, much research evaluating the radiographic destruction of finger joints in patients with rheumatoid arthritis (RA) using deep learning models was conducted. Unfortunately, most previous models were not clinically applicable due to the small object regions as well as the close spatial relationship. In recent years, a new network structure called RetinaNets, in combination with the Focalloss function, proved reliable for detecting even small objects. Based on this, the authors aimed to increase the recognition performance to a clinically valuable level by proposing an innovative approach with adaptive changes in intersection over union (IoU) values during training of our Retina Networks using the focal loss error function. To this end, the authors retrospectively analyzed 300 conventional radiographs from 119 RA patients and determined the erosion score according to the Sharp van der Heijde (SvH) metric.Subsequently, they trained a standard RetinaNet with different IoU values as well as adaptively modified IoU values and compared them in terms of accuracy, mean average accuracy (mAP), and IoU. The authors demonstrated that their approach, improved the erosion detection accuracy up to 94% and an mAP of 0.81 ± 0.18. In contrast with static IoU values, they only achieved an accuracy of 80% and an mAP of 0.43 ± 0.24. They concluded that the adaptive adjustment of IoU values during training is a simple and effective method to increase the recognition accuracy of small objects such as finger and wrist joints. Innovative, interesting, and well written manuscript. I have some minor suggestions with a pure academic spirit.”
Authors’ Response: We want to thank the reviewer for taking the time to review our manuscript and the general appreciation and insightful and constructive comments. Please read our response to each comment below, where we address each comment separately.
R2 Comment #1: “I suggest to be more explicit in the purpose “Therefore, our study aimed to investigate how the IoU threshold affects classification accuracy and whether a novel approach we proposed using adaptive IoU thresholds would provide higher detection and localization accuracy.””
Authors’ Response: We agree with the reviewer that the current wording regarding the purpose of the study is not precisely worded and is difficult to read.
Authors‘ Action: In light of this validated comment, we have reworded the purpose of this study. The revised section reads:
“Therefore, the study aimed to investigate whether a novel approach based on adaptive IoU thresholds would provide better localization and detection accuracy. “ (lines 75f)
R2 Comment #2: “Describe figure 1 in the body of the manuscript. Furthermore, the legend must explain the six parts of the figure.”
Authors’ Response: This is another excellent commentary supporting the manuscript’s readability.
Authors‘ Action: In accordance with the comment, we have revised the Figure Caption and described the figure in the Main Text. The correspondingly revised Figure Caption and passage in the Main Text read:
“For all images, P.S., an experienced rheumatologist (15 years of experience in Rheumatology, including clinical trials and scoring Sharp van der Heijde (SvH) method), determined the erosion score according to van der Heijde et al. [24,25] assigning a value between 0 and 5 to each joint depending on the extent of erosion (Figure 1).” (lines 94ff)
R2 Comment #3: “Check the resolution of the figure.”
Authors’ Response: We would like to thank the reviewer for thoroughly reviewing the manuscript and checking the resolution of the figures. The poorer resolution in the Word document resulted from image compression within Word.
Authors‘ Action: The poorer resolution in the Word document resulted from image compression within Word. In the revised manuscript, you will now find the figures without compression in the correct resolution.
R2 Comment #4: “What about inserting a table/list with the acronyms?”
Authors’ Response: We agree with the reviewer that a listing of abbreviations and acronyms is helpful for readability and understanding.
Authors‘ Action: At the reviewer's suggestion, we have included such a listing at the end of the manuscript. The list of abbreviations and acronyms is as follows:
“Abbreviations and acronyms
SvH – Sharp van der Heijde
CEST - Chemical Exchange Saturation Transfer
dGEMRIC - delayed gadolinium-enhanced MRI of cartilage
DL – Deep Learning
RetinaNet – Retina networks
IoU - Intersection over Union
GT - ground truth
CR - conventional radiographs
PACS - Picture Archiving and Communication System
DICOM - Digital Imaging and Communications in Medicine
mAP - mean average accuracy
ReLU - Rectified Linear Unit
ResNet - residual neural network
FPN - feature pyramid network
FNN - feedforward neural networks
YOLO - You only look once
SSD - single-shot detector” (lines 442ff)
R2 Comment #5: “Avoid the use of we/our”
Authors’ Response and Action: We would like to thank the reviewer for this further excellent comment. As pointed out by the reviewer, we have reduced the use of "we" and "our" in the manuscript. Due to the large number of revised sections, the following is only a selection of revised passages:
“Therefore, the study aimed to increase the recognition performance to a clinically valuable level by proposing an innovative approach with adaptive changes in intersection over union (IoU) values during training of Retina Networks using the focal loss error function.” (lines 75f)
“In contrast Retina networks with static IoU values achieved only an accuracy of 80% and an mAP of 0.43 ± 0.24.” (lines 33f)
“In order to generate a large amount of artificial data during training, extensive data augmentation was used.” (lines 144f)
R2 Comment #6: “29 \ 54” Usually the slash is “/”
Authors’ Response: We would like to thank the reviewer for thoroughly reviewing the manuscript.
Authors‘ Action: In accordance with the reviewer's comment, we have changed the slash. In addition, as suggested by Reviewer 1, the table has been moved from the Results section to the Material & Methods section. The revised table reads as follows:
|
parameters |
Overall |
Training |
Validation |
Test |
|
Age [a] |
55.6 ± 12.2 (23 – 87) |
55 ± 13 (23 – 87) |
56 ± 7 (29 – 82) |
57 ± 12 (27 – 87) |
|
Number patients |
119 |
83 |
12 |
24 |
|
Number images |
300 |
231 |
45 |
24 |
|
male / female |
41 / 78 |
29 / 54 |
4 / 8 |
6 / 18 |
|
Sum of erosion score |
52 ± 22 |
53 ± 25 (32 – 179) |
46 ± 8 (32 – 101) |
55 ± 30 (32 – 179) |
|
Mean erosion score |
1.52 ± 0.70 (1.0 – 5.59) |
1.65 ± 0.77 (1.0 – 5.59) |
1.43 ± 0.25 (1.0 – 4.3) |
1.72 ± 0.95 (1 – 5.59) |
|
Min erosion score |
0.0 ± 0.0 (0 - 0) |
0.0 ± 0.0 (0 - 0) |
0.0 ± 0.0 (0 – 0) |
0.0 ± 0.0 (0-0) |
|
Max erosion score |
4.1 ± 1.92 (1 – 6) |
4.2 ± 1.87 (1 – 6) |
3.2 ± 1.7 (1 – 6) |
4.6 ± 1.6 (1 – 6) |
R2 Comment #7: „Check if equations (1-3) follow the MDPI standards“
Authors’ Response: We thank the reviewer for his careful review of the manuscript and for pointing out the style guidelines of MDPI to keep in mind. We checked the equations, and they were in accordance with MDPI standards.
Round 2
Reviewer 1 Report
accept